# Random smooth gray value transformations for cross modality learning with gray value invariant networks

## Abstract

Random transformations are commonly used for augmentation of the training data with the goal of reducing the uniformity of the training samples. These transformations normally aim at variations that can be expected in images from the same modality. Here, we propose a simple method for transforming the gray values of an image with the goal of reducing cross modality differences. This approach enables segmentation of the lumbar vertebral bodies in CT images using a network trained exclusively with MR images.

## 1. Introduction

Detection and segmentation networks are typically trained for a specific type of images, for instance MR images. Networks that reliably recognize an anatomical structure in those images most often completely fail to recognize the same structure in images from another imaging modality. However, a lot of structures arguably *look* similar across modalities. We therefore hypothesize that a network could recognize those structures if the network was not specialized to the gray value distribution of the training images but was invariant to absolute and relative gray value variations.

Invariance to certain aspects of the data is commonly enforced by applying random transformations to the training data, such as random rotations to enforce rotational invariance, or similar transformations (Roth et al., 2016; Khalili et al., 2019). In this paper, we present a simple method for randomly transforming the gray values of an image while retaining most of the information in the image. We demonstrate that this techniques enables cross modality learning by training a previously published method for segmentation of the vertebral bodies (Anonymous) with a set of MR images and evaluating its segmentation performance on a set of CT images.

## 2. Method

We define a transformation function $y(x)$ that maps a gray value $x$ to a new value. This function is a sum of $N$ sines with random frequencies, amplitudes and offsets. A sum of sines is a straightforward way of creating a smooth and continuous but non-linear transformation function. A continuous function ensures that gray values that are similar in the original image are also similar in the transformed image so that homogeneous structures remain homogeneous despite the presence of noise.

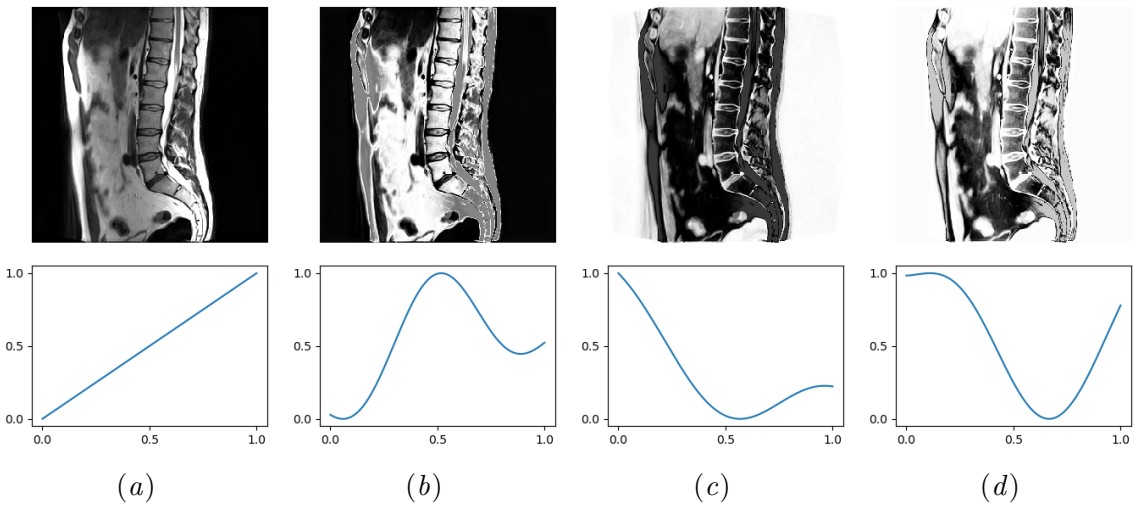

Figure 1: T2-weighted lumbar spine MRI scan, (a) the original image slice and (b)–(d) the same slice after applying the proposed random smooth gray value transformations. The corresponding randomly generated transformation function is shown below each image.

The transformation function is therefore defined as:

$$y(x) = \sum_{i=1}^{N} A_i \cdot \sin(f_i \cdot (2\pi \cdot x + \varphi_i)). \tag{1}$$

The frequencies $f_i$ are uniformly sampled from $[f_{\min}, f_{\max}]$. This range of permitted frequencies and the number of sines $N$ determine the aggressiveness of the transformation, i.e., how much it deviates from a simple linear mapping. The amplitudes $A_i$ are uniformly sampled from $[-1/f, 1/f]$, which ensures that low frequency sines dominate so that the transformation function is overall relatively calm. The offsets $\varphi_i$ are uniformly sampled from $[0, 2\pi]$. A few examples of randomly transformed MR images are shown in Figure 1.

For simplicity, we fix the input range of the neural network to values in the range $[0, 1]$. All input images regardless of modality need to be normalized to this range during training and inference. CT images were normalized by mapping the fixed range $[-1000, 3000]$ to $[0, 1]$, for MR images this range was based on the 5 % and 95 % percentiles of the image values. Values below 0 or above 1 were clipped. During training, the images were additionally randomly transformed according to Equation 1 and again scaled to $[0, 1]$.

## 3. Results

To evaluate the cross modality learning claim, we trained a method for segmentation of the vertebral bodies (Anonymous) with a set of T2-weighted lumbar spine MRI scans. These scans and corresponding reference segmentations of the vertebral bodies were collected

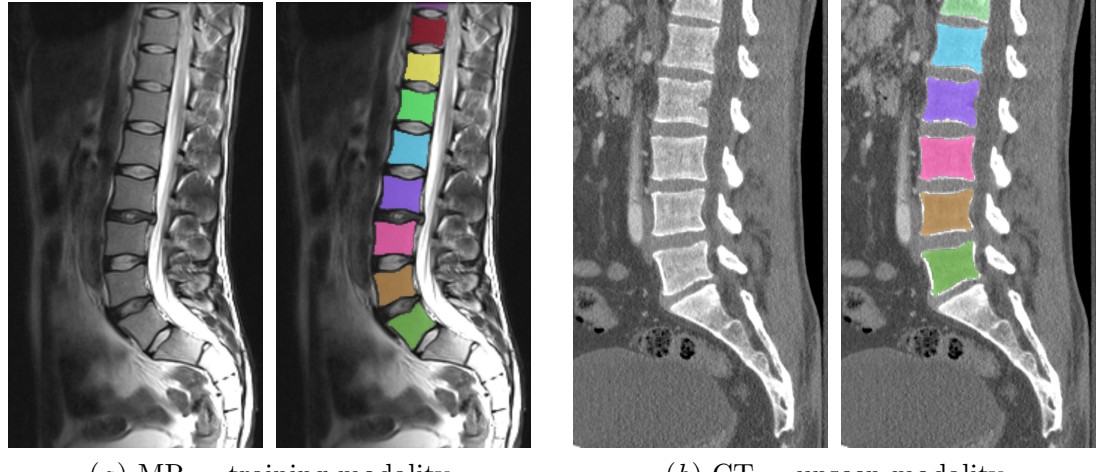

(*a*) MR = training modality        (*b*) CT = unseen modality

Figure 2: Segmentation results for (a) an image from the training modality, and (b) an image from another modality in which the target structure (the vertebral bodies) has somewhat similar appearance. Note that these are the results when training the segmentation network with MR images to which random smooth gray value transformations were applied during training.

from two publicly available datasets (Chu et al., 2015; Zukić et al., 2014). We used in total 30 scans for training, 3 for validation and kept 7 scans for testing. Additionally, we collected a second test set consisting of 3 contrast-enhanced CT scans from another publicly available dataset (Yao et al., 2016). This dataset contains reference segmentations of the entire vertebrae, and the images show all thoracic and lumbar vertebrae. We therefore first cropped the images to the lumbar region and then manually removed the posterior parts of the vertebrae from the masks to create reference segmentations of the vertebral bodies.

Three experiments were performed: (1) The segmentation network was trained without any gray value modifications other than normalization to the $[0, 1]$ input range. (2) The segmentation network was trained with randomly inverted gray values so that dark structures like bone appear bright in every other training sample and vice versa. (3) The segmentation network was trained with the proposed random smooth gray value transformations applied to the training samples. We used frequencies from $f_{\min} = 0.2$ to $f_{\max} = 1.6$, and the transformation functions were sums of $N = 4$ sine functions.

For the MR test set, the Dice scores were virtually identical in all three experiments, the addition of the data augmentation step did neither negatively nor positively influence the segmentation performance (1: $91.7 \pm 3.1\,\%$, 2: $92.6 \pm 1.7\,\%$, 3: $91.9 \pm 2.2\,\%$). For the CT test set, the segmentation was only successful when random smooth gray value transformations were applied during training (1: $53.2 \pm 29.5\,\%$, 2: $75.6 \pm 26.7\,\%$, 3: $94.1 \pm 1.1\,\%$). Examples are shown in Figure 2. These results demonstrate that the proposed gray value transformations can enable training of gray value invariant networks.

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
