# OpenReview forum: "Random smooth gray value transformations for cross modality learning with gray value invariant networks"
_MIDL.io/2020/Conference — Submitted to MIDL 2020_

### Official Review · AnonReviewer4 · 2020-03-08
**This paper addresses the problem of multimodal lumbar vertebral body segmentation by training the networks to be invariant to gray values.**

**Rating:** 2
**Confidence:** 5

**Review:**

The paper proposes a gray value augmentation on data, to make the networks invariant to image intensities. The problem statement is clearly described. However, some important details about the segmentation network, training scheme and labels are missing.

Strength:
The proposed idea for cross modality segmentation is novel and reported dice scores are fare.

Weakness:
- The model is validated on a few number of CT images, so the results are not necessarily generalizable.
- The choice of segmentation network architecture and also the training scheme is not reported.
- It is not clear whether the network is predicting a binary mask for all vertebrae or it is predicting a unique label for each vertebrae. This effects the reported dice scores considerably.
- If the segmentation is multi-label (according to Fig. 2), How many labels were used to train the network?
There should be 5 (for some cases 6) labels defined for lumbar vertebrae, How does the network handle the thoracic vertebrae visible in the Fig. 2?

I think another possible validation could be comparing the histograms of intensities in the training dataset (after augmentation) and target test set.

---

### Official Review · AnonReviewer2 · 2020-03-13
**a simple approach that probably only works in the proposed scenario**

**Rating:** 1
**Confidence:** 5

**Review:**

It is refreshing to read a paper claiming that all that you need for cross-modality learning is good data augmentation. However, modality transfer should be approached cautiously since different image modalities of the same anatomical sites are used to capture complementary information. Thus, modality transfer should only be used when aiming for information available in both modalities. In the case of vertebrae, cortical bone clearly visible as the brightest structure in the CT images can be easily mixed with cartilage surrounding vertebrae that can be seen in MR images as dark structure surrounding the vertebral body. This is probably the reason why there is a clear under-sampling segmentation in CT (check the Fig. 2). Moreover, the proposed augmentation looks too simplified to be used in any other scenarios except the one proposed in the manuscript. Lumbar vertebrae are well-separated bones in CT that are not difficult to segment also with a clever chose of a threshold-based method. I doubt that going for thoracic vertebrae or any other more demanding anatomy the proposed approach would give a decent result. Moreover, compare to modality transfer from MR to CT, the opposite direction from CT to MR is a way more challenging, but also more realistic scenario, since having a bone labels in CT images are more commonly available. It is also not clear why authors used a sum of sines to create non-linear transformation function and not e.g. a polynomial function of the third degree.

---

### Official Review · AnonReviewer1 · 2020-03-13
**Simple method for cross-modality learning, seems to be effective, limited experiments and no comparison with alternative methods**

**Rating:** 2
**Confidence:** 4

**Review:**

The authors propose a method to train gray-value-invariant networks by applying random gray value transformations to the images during training. The paper proposes an intensity transformation using sine functions with random parameters. The method is evaluated on an MR-to-CT lumbar segmentation problem, showing that it improved cross-modality learning.

Strengths:
The proposed intensity transformation is simple and easy to apply. Based on the (limited) experiments in the paper, the method does seem to improve the cross-domain classification in this case. It is also useful to see that the augmentation did not hurt the performance in the original MRI domain.

Weaknesses:
There is no comparison with alternative methods. The paper's assertion that having smooth gray-value transformations is better than having non-smooth transformations is difficult to evaluate, because there are no results for other ways to train gray-value-invariant methods.

The experiments are limited in other ways as well: the dataset is small and there is no information about the network or how this was trained or evaluated. (And there is still one half page left where this could have been explained.)

The results on page 3 would have looked better in a table.

---

### Official Review · AnonReviewer3 · 2020-03-13
**This short paper introduces a transformation that modifies gray-level values within grayscale training images to enable cross-modality detection for vertebral bodies from MRI images to CT images.**

**Rating:** 2
**Confidence:** 4

**Review:**

The methodology introduced within the paper is clear, simple and concise. The authors propose combining N=4 sine wave frequencies to apply a continuous transformation function that modifies the gray values of the training dataset. The transformation preserves edge gray-value information and is effective in assisting vertebral body segmentation.

Cons: The authors fail to acknowledge or mention whether this transformation is applicable to other modalities (for example: T1-weighted MRI input data, or CT to MRI detection) and other less-bony structures.

Potential impact: Preliminary results demonstrate that the transformation enhances and preserves, the vertebral body edges that leads to good cross-modality segmentation. I think the introduced transformation has potential use and impact for generalizing datasets during training where structures being segmented have edges to guide localization/segementation. It would be interesting to see whether the introduced transformation can assist with cross-modality whole-vertebra and intervertebral disc segmentation, where there is low bone/tissue image intensity contrast.

---

### Meta-Review · Area_Chair1 · 2020-04-06
**MetaReview of Paper248 by AreaChair1**

**Rating:** 2

**Metareview:**

This paper proposes an intensity transformation method from MR to CT. The reviewers have major concerns on method evaluation, which only involves a few number of CT images and has no comparison with alternative methods; method setting, e.g., number of labels, modality transfer directions (MR to CT, but not CT to MR), network and training setting; and ability to generalize to other regions or modalities.

**Paper Type:**

methodological development

---

### Decision · Program_Chairs · 2020-04-11

Reject